# Prediction of All-Cause Mortality Following Percutaneous Coronary Intervention in Bifurcation Lesions Using Machine Learning Algorithms

**DOI:** 10.3390/jpm12060990

**Published:** 2022-06-17

**Authors:** Jacopo Burrello, Guglielmo Gallone, Alessio Burrello, Daniele Jahier Pagliari, Eline H. Ploumen, Mario Iannaccone, Leonardo De Luca, Paolo Zocca, Giuseppe Patti, Enrico Cerrato, Wojciech Wojakowski, Giuseppe Venuti, Ovidio De Filippo, Alessio Mattesini, Nicola Ryan, Gérard Helft, Saverio Muscoli, Jing Kan, Imad Sheiban, Radoslaw Parma, Daniela Trabattoni, Massimo Giammaria, Alessandra Truffa, Francesco Piroli, Yoichi Imori, Bernardo Cortese, Pierluigi Omedè, Federico Conrotto, Shao-Liang Chen, Javier Escaned, Rosaly A. Buiten, Clemens Von Birgelen, Paolo Mulatero, Gaetano Maria De Ferrari, Silvia Monticone, Fabrizio D’Ascenzo

**Affiliations:** 1Division of Internal Medicine and Hypertension, Department of Medical Sciences, University of Turin, 10126 Turin, Italy; jacopo.burrello@unito.it (J.B.); paolo.mulatero@libero.it (P.M.); silvia.monticone@unito.it (S.M.); 2Division of Cardiology, Department of Medical Sciences, University of Turin, 10126 Turin, Italy; guglielmo.gallone@gmail.com (G.G.); ovidio.defilippo@gmail.com (O.D.F.); francescopiroli@yahoo.it (F.P.); pierluigiomede@gmail.com (P.O.); federico.conrotto@gmail.com (F.C.); fabrizio.dascenzo@gmail.com (F.D.); 3Department of Electrical, Electronic and Information Engineering, University of Bologna, 40126 Bologna, Italy; alessio.burrello@unibo.it; 4Department of Control and Computer Engineering, Polytechnic University of Turin, 10129 Turin, Italy; daniele.jahier@polito.it; 5Cardiology Department, Medisch Spectrum Twente, Thoraxcentrum Twente, 7412 Enschede, The Netherlands; e.ploumen@mst.nl (E.H.P.); g.zocca@mst.nl (P.Z.); r.buiten@mst.nl (R.A.B.); c.vonbirgelen@mst.nl (C.V.B.); 6Cardiology Department, San Giovanni Bosco Hospital, 10154 Turin, Italy; mario.iannaccone@hotmail.it; 7Division of Cardiology, S. Giovanni Evangelista Hospital, Tivoli, 00019 Rome, Italy; leo.deluca@libero.it; 8Coronary Care Unit and Catheterization Laboratory, A.O.U. Maggiore della Carità, 28100 Novara, Italy; giuseppe.patti@uniupo.it; 9Department of Cardiology, San Luigi Gonzaga Hospital, 10043 Orbassano, Italy; enrico.cerrato@gmail.com; 10Department of Cardiology, Medical University of Silesia, 40-752 Katowice, Poland; wojtek.wojakowski@gmail.com; 11Division of Cardiology, A.O.U. “Policlinico-Vittorio Emanuele”, 95123 Catania, Italy; giuseppevenuti@live.it; 12Structural Interventional Cardiology, Careggi University Hospital, 50134 Florence, Italy; amattesini@gmail.com; 13Department of Cardiology, Aberdeen Royal Infirmary, Aberdeen AB25 2ZN, UK; nicolaryan@gmail.com; 14Department of Cardiology, Pierre and Marie Curie University, 75005 Paris, France; gerald.helft@aphp.fr; 15Department of Medicine, Università degli Studi di Roma Tor Vergata, 00133 Rome, Italy; saveriomuscoli@gmail.com; 16Division of Cardiology, Nanjing First Hospital, Nanjing Medical University, Nanjing 210029, China; jkan@126.com; 17Division of Cardiology, Pederzoli Hospital, 37019 Peschiera del Garda, Italy; isheiban@gmail.com; 18Department of Cardiology, University Clinical Hospital, 02-091 Warsaw, Poland; radoslaw.parma@gmail.com; 19Department of Cardiovascular Sciences, IRCCS Centro Cardiologico Monzino, 20138 Milan, Italy; daniela.trabattoni@cardiologicomonzino.it; 20Division of Cardiology, Ospedale Maria Vittoria, 10144 Turin, Italy; massimo.giammaria@aslcittaditorino.it; 21Division of Cardiology, ASL Cardinal Massaia Hospital, 14100 Asti, Italy; truffaalessandra@yahoo.it; 22Department of Cardiovascular Medicine, Nippon Medical School, Sendagi, Bunkyo-ku, Tokyo 113-8602, Japan; yoichiimori@me.com; 23Division of Cardiology, San Carlo Clinic, 20037 Milan, Italy; bcortese@gmail.com; 24Division of Cardiology, Hospital San Carlos, Complutense University, 28040 Madrid, Spain; escaned@secardiologia.es

**Keywords:** percutaneous coronary intervention, coronary bifurcation, machine learning, prognosis

## Abstract

Stratifying prognosis following coronary bifurcation percutaneous coronary intervention (PCI) is an unmet clinical need that may be fulfilled through the adoption of machine learning (ML) algorithms to refine outcome predictions. We sought to develop an ML-based risk stratification model built on clinical, anatomical, and procedural features to predict all-cause mortality following contemporary bifurcation PCI. Multiple ML models to predict all-cause mortality were tested on a cohort of 2393 patients (training, n = 1795; internal validation, n = 598) undergoing bifurcation PCI with contemporary stents from the real-world RAIN registry. Twenty-five commonly available patient-/lesion-related features were selected to train ML models. The best model was validated in an external cohort of 1701 patients undergoing bifurcation PCI from the DUTCH PEERS and BIO-RESORT trial cohorts. At ROC curves, the AUC for the prediction of 2-year mortality was 0.79 (0.74–0.83) in the overall population, 0.74 (0.62–0.85) at internal validation and 0.71 (0.62–0.79) at external validation. Performance at risk ranking analysis, k-center cross-validation, and continual learning confirmed the generalizability of the models, also available as an online interface. The RAIN-ML prediction model represents the first tool combining clinical, anatomical, and procedural features to predict all-cause mortality among patients undergoing contemporary bifurcation PCI with reliable performance.

## 1. Introduction

The evolution of both stent technology and implantation techniques has translated into improved clinical outcomes following percutaneous coronary intervention (PCI) in complex anatomical and procedural settings, such as coronary bifurcation lesions [1,2,3,4,5].

However, real-life contemporary registries [6,7] still report a considerable risk of adverse events in this high-risk subset, warranting precise prognostication.

Available risk scores to predict adverse events associated with PCI are based on study populations with a small proportion of bifurcation lesions [8,9,10,11]. The external performance of such models in this lesion setting remains modest [12]. The absence of dedicated algorithms to predict long-term outcomes of PCI in coronary bifurcations clashes with the abundant evidence demonstrating a significantly poorer short- and long-term prognosis of these lesions compared to the overall PCI population, most likely as a result of multifaceted differences in procedural technique, hemorheology and vessel healing that makes PCI in bifurcations less forgiving than in other anatomical subsets [5,13,14,15,16]. While the risk stratification of ischemic endpoints is pivotal to informing patient management and therapeutic choices, mortality prediction remains an important goal to improve physician-patient communication, orientate follow-up and clinical decision making, and allow comparative effectiveness research in order to guide procedural strategy and further technical advances [17,18]. However, to date, there is no available predictive tool to predict long-term mortality following bifurcation PCI.

In the clinical research field of risk prediction following PCI, available studies favored either clinical [8] or anatomy/procedure [11,19] focused approaches to assess residual risk rather than exploiting the multidimensional nature of risk, which may be better determined by integrating these factors, especially in the bifurcation setting. Moreover, traditional prognostic risk assessment is constructed upon a limited selection of variables based on a priori assumptions, potentially omitting routinely assessed, powerful outcome predictors. This potential limit of classical inferential statistics could be overcome by machine learning (ML) that adopts a radically different approach, focusing on algorithmic representations of data and their classification in order to establish and quantify the relationships among variables [20]. Indeed, ML has emerged as a powerful approach to circumvent the limitations of current methods by applying computational algorithms to large datasets with numerous multiparametric variables, capturing high-dimensional, non-linear relationships among clinical features to make data-driven outcome predictions [21]. The effectiveness of this strategy has been shown across several cardiovascular applications, where ML was superior to validated traditional risk stratification tools, including the prediction of adverse events among patients with coronary artery disease or heart failure undergoing cardiac resynchronization therapy [22,23,24]. Thus, we developed an ML-based risk stratification model integrating clinical, anatomical, and procedural features to predict mortality following bifurcation PCI, utilizing a large international cohort of patients undergoing coronary bifurcation PCI with contemporary stents [5]. The model was then validated in a large population derived from two randomized trials of contemporary stents.

## 2. Materials and Methods

### 2.1. Study Population

The present study includes 4094 patients with a coronary bifurcation lesion treated with contemporary, very thin drug-eluting stents. The ML-based model was developed and internally validated using the RAIN (very thin stents for patients with left main or bifurcation in real life, NCT03544294) registry population (defined as the discovery cohort in the present study). RAIN is a multicenter retrospective registry including consecutive patients undergoing unprotected left main or coronary bifurcation PCI from 2014 to 2017 at 23 institutions worldwide [5]. Patients undergoing ostial/mid-shaft left main PCI or patients with incomplete clinical, angiographic, procedural, and outcome data were excluded from this study. Thus, 2393 patients undergoing bifurcation PCI with very thin strut stents comprised the final discovery cohort. The discovery cohort was randomly divided into a training cohort (n = 1795) and an internal validation cohort (n = 598).

The external validation cohort for the ML-based model was obtained from the merged DUTCH PEERS (Durable polymer-based stent Challenge of Promus ElemEnt versus Resolute integrity: TWENTE II) trial and BIO-RESORT trial patient cohorts [2,25,26]. More specifically, 1701 patients with a coronary bifurcation lesion treated with very thin stents (n = 465 from DUTCH-PEERS, n = 1236 from BIO-RESORT) with follow-up truncated at 2-years comprised the external validation cohort.

Detailed descriptions of the study cohorts are provided as Appendix A. Cardiovascular risk factors, clinical presentation, angiographic features, use of intravascular imaging, bifurcation technique details, characteristics of the treated lesion, and implanted stents were collected in a dedicated database. The primary endpoint of the study was all-cause mortality at two years, while all-cause death at 30 days and at one year were evaluated as secondary endpoints. The study complies with the Declaration of Helsinki, all of the patients provided informed consent for inclusion in the registries, and local institutional review board approval was obtained by each center.

### 2.2. Model Development

An overview of model development is provided in Figure 1A. The model was trained and validated according to TRIPOD guidelines. The discovery cohort was randomly split into a training (n = 1795) and an internal validation (n = 598) dataset. The Fisher score was used for feature selection in the training cohort, and the variables with a coefficient >0.75 were retained (Appendix A); the selected variables were used to develop predictive models. A grid search including 5 different ML classifiers (linear discriminant analysis [LDA], random forest [RF] regressor, support vector machine [SVM] with a linear or Gaussian kernel, and isolation forest) and 3 algorithms for data imbalance correction (synthetic minority over-sampling technique [SMOTE], SMOTE and nearest neighbors, and random oversampling methods) was performed on the training cohort, generating 13 models for mortality prediction after bifurcation PCI. Data imbalance correction algorithms were applied to avoid the accuracy paradox (a falsely high accuracy due to over-prediction of the most represented class); oversampling techniques impute simulated patient data starting from real patients from the discovery cohort in the virtual space created by patient parameters to balance the number of patients with death occurrence and patients without events during model training. LDA applies a linear combination approach; the predicted endpoint is derived from the following equation: “Endpoint (all-cause mortality) = LDAcoeff_1_*Variable_1_ + LDAcoeff_2_*Variable_2_ + … + LDAcoeff_n_*Variable_n_ > tested thresholds”. The coefficients are generated by the algorithm to maximize the separation between groups (Death vs. No events), increasing precision estimates by variance reduction; variables represent patients’ features, selected as described above. The RF algorithm generates a pre-defined set of classification trees (“n” classification trees) with a fixed maximum number of splits for each tree. The predicted endpoint results from the outcome of each classification tree of the forest; if at least “(n/2) + 1” of “n” trees of the RF predicts death as an outcome, then this endpoint is assigned to the patient. Linear SVM builds a classification model to assign patients to their outcome given a linear boundary, while Gaussian SVM allows the patients to be divided using a non-linear boundary. The model equations are: “SVMcoeff_0_ + SVMcoeff_1_*Variable_1_ + SVMcoeff_2_*variable_2_ + …. + SVMcoeff_n_*Variable_n_”, and “SVMcoeff_0_ + SVMcoeff_1_*f(Variable_1_) + SVMcoeff_2_*f(variable_2_) + …. + SVMcoeff_n_*f(Variable_n_)”, respectively, where “f” is an exponential function coefficient. Isolation forest is a particular type of RF that uses unsupervised learning to discriminate anomalies (in this case, patients with death occurrence) from normal data (patients without events).

A random forest regressor algorithm with random oversampling correction yielded the highest accuracy for the prediction of death occurrence, and it is referenced throughout the manuscript as the RAIN-ML prediction model. A 10-fold cross-validation was used to select and tune the hyper-parameters (number of classification trees and number of splits) of the RAIN-ML model in the training cohort; the hyper-parameters reaching the highest accuracy in outcome prediction were selected. Thereafter, its performance was assessed by K-center cross-validation, risk stratification analysis, continual learning, and both internal and external validation. Overfitting bias was defined as the difference between the accuracy obtained during training and the accuracy during the internal or external validation. The model was developed to predict 2-years all-cause mortality; its performance was then assessed also at different time points (30-day and 1-year). A detailed description of the model development is provided in the Appendix A (Extended Methods section).

A user-friendly online interface was designed to facilitate the application of the RAIN-ML prediction model in clinical practice (available at https://rain.hpc4ai.it; accessed on 12 May 2022).

### 2.3. Statistical Analysis

Categorical variables were reported as count and percentage and analyzed by chi-square test. Continuous variables were reported as median and interquartile ranges and analyzed by Mann–Whitney U-test. The analysis of the receiver operating characteristic (ROC) curves was performed to calculate the area under the curve (AUC) and to derive the best cut-off by evaluation of the Youden Index (J = sensitivity + specificity − 1). A two-tailed *p*-value of less than 0.05 was considered significant. Analyses were performed by IBM SPSS Statistics 26 (IBM, New York, NY, USA), Python 3.5 (library, scikit-learn), and GraphPad PRISM 8.0 (La Jolla, California, CA, USA).

## 3. Results

### 3.1. Characteristics of the Discovery Cohort

The discovery cohort (n = 2393) was used to develop and internally validate the RAIN-ML prediction model. The baseline characteristics of the patients undergoing bifurcation PCI (median age 69 [interquartile range, IQR: 61–77] years, male sex 76.0%) from the discovery cohort are reported in Appendix A, as stratified by death occurrence. The discovery cohort was randomized into a training and an internal validation dataset to develop and test the predictive models (Table 1). There were no differences between the training and internal validation cohorts (Appendix A). After a median follow-up of 274 (IQR 52–434) days, 137 (5.7%) patients died (103 and 34 from the training and internal validation cohort, respectively; 30-day, 1-year, and 2-year mortalities were 1.2%, 3.7%, and 5.2%, respectively).

### 3.2. Development and Internal Validation of the RAIN-ML Prediction Model

Several patient and lesion-related parameters differed significantly between patients in whom the all-cause death endpoint was or was not reached at 2-year follow-up (Appendix A). Features associated with all-cause mortality were selected by Fisher score (see Methods and Extended Methods sections) in the training cohort. Of the 38 patient and lesion-related parameters, 13 were excluded leading to a final set of 25 input variables (13 related to patient history and clinical presentation, seven to coronary anatomy, and five to the PCI procedure; Appendix A). Chronic kidney disease (CKD, defined as a glomerular filtration rate < 60 mL/min/1.73 m^2^) was the best predictor of all-cause mortality, followed by the indication for PCI, first lesion vessel, diabetes, diffuse coronary disease, left ventricular ejection fraction (EF, %), kind of bifurcation, and age (Figure 1B).

Among the 13 trained models, a random forest regressor algorithm with random oversampling correction yielded the highest accuracy in predicting 2-years all-cause mortality and was selected as the RAIN-ML prediction model (Appendix A). A representative classification tree of the random forest RAIN-ML model is shown in Figure 1C. After tuning (Appendix A), the RAIN-ML model displayed an accuracy of 81.1% at training (AUC 0.791; 95% CI 0.742–0.840), and 79.8% at internal validation (AUC 0.768; 95% CI 0.669–0.868), with an overfitting effect of 1.3% (Figure 1D). The sensitivity and specificity were 82.5/81.0% during training and 67.6/80.5% during internal validation, with 85 of 103 and 23 of 34 patients experiencing the correctly classified endpoint.

The performance of the RAIN-ML model in all-cause mortality prediction was assessed at different time points: the model was developed to predict 2-year all-cause mortality as the primary endpoint, and then its performance was also assessed at 30-days follow-up, 1-year follow-up, and including all events; Figure 2). After 1 year, the AUC was 0.777 (95% CI 0.721–0.834) during training and 0.718 (95% CI 0.586–0.850) during internal validation (Figure 2B). After 2 years, the AUC was 0.799 (95% CI 0.745–0.852) during training and 0.736 (95% CI 0.624–0.847) during internal validation (Figure 2C).

To further confirm the generalizability of the RAIN-ML prediction model, we applied a K-center cross-validation approach to the discovery cohort (n = 2393). The analysis confirmed an acceptable performance in each of the 23 participating institutions, with a mean accuracy, sensitivity, and specificity of 75.3%, 60.6%, and 76.2%.

### 3.3. External Validation of the RAIN-ML Model

The patients from the external validation cohort were younger (65 [IQR 57–72] years), with a lower prevalence of cardiovascular risk factors, prior myocardial infarction, and coronary revascularizations compared to the discovery cohort (Table 1 and Appendix A). At external validation, 1312 of 1701 patients were correctly classified according to death occurrence (Figure 1D), resulting in an accuracy of 77.1% (overfitting effect 4%). ROC curve analysis confirmed a good performance at all the evaluated times of follow-up. After 2 years, 39 patients died (2.3%), and the AUC was 0.706 (95% CI 0.619–0.794; Figure 2C). The predictive performance in the overall population (mixed discovery and external validation cohort; n = 4094) was similar, with an AUC of 0.769 (95% CI 0.728–0.810), 0.726 (95% CI 0.633–0.819), 0.758 (95% CI 0.707–0.810), and 0.786 (95% CI 0.743–0.830), respectively, considering all events, or a follow up of 30 days, 1 or 2 years (Figure 2).

### 3.4. Risk Stratification Analysis

In the mixed discovery and external validation cohorts, increasing coefficients of the RAIN-ML prediction model were directly correlated with the proportion of subjects with death occurrence (Figure 3A–D). Patient stratification according to the RAIN-ML model and the occurrence of death at follow-up are reported in Appendix A. The lowest risk patients with an ML model coefficient of 0.10–0.19 displayed an all-cause mortality risk of 0.3%, 0.9%, and 2.3%, after 30 days, 1-year, and 2-year follow-up, respectively. On the other hand, the highest risk patients with an ML model coefficient of 0.90–1.00 displayed an all-cause mortality risk of 8.1%, 23.7%, and 72.2%, respectively, after 30 days, 1-year, and 2-year follow-up. Using cut-offs derived by ROC curve analysis to optimize sensitivity and specificity, we then stratified patients according to the predicted risk of all-cause mortality. For the RAIN-ML prediction model, a coefficient of less than 0.21 identified a low-risk subgroup of patients with a risk of 1.4% of all-cause mortality (35 of 2444 subjects), patients with a coefficient ranging between 0.21 and 0.70 showed an intermediate risk of 4.5% (52 of 1166 subjects), while a coefficient higher than 0.70 identified a risk of 18.4% (high-risk group; 89 of 484 subjects; Figure 3E). As compared to low risk, being categorized as intermediate risk and high risk was associated with increased (3.2-fold and 13.1-fold, respectively, both *p* < 0.001) mortality. The risk ranking approach (after the exclusion of patients at intermediate risk) led to a sensitivity/specificity of 71.8/85.9% with an overall accuracy of 85.3% when low-risk patients were compared to those classified as high risk.

The predictive performances of the RAIN-ML model are summarized in Appendix A.

## 4. Discussion

A reliable and clinically relevant patient risk stratification is a prerequisite for adequate treatment selection, informed consent, and improved care, all key elements of modern personalized medicine. Following this guiding principle, we developed and validated a prediction model based on supervised machine learning algorithms to identify long-term all-cause mortality in patients undergoing PCI on coronary bifurcations.

Our model was first developed using the largest available bifurcation PCI registry reflective of contemporary practice, encompassing a wide range of very thin second-generation drug-eluting stents and procedural techniques, applied to a variety of clinical scenarios among all-comers at 23 institutions worldwide. Subsequently, our findings were externally validated using a large bifurcation PCI population derived from two randomized trials of second-generation drug-eluting stents [2,25,26].

The RAIN-ML model, correctly classifying 3245 of 4094 patients, showed a good discriminative capability for all-cause mortality prediction, confirmed at both internal and external validation, and by K-center cross-validation, with an accuracy of 81.1% during training and ranging between 77.1% and 79.8% during validation. According to the RAIN-ML prediction model, following bifurcation PCI, 6% of the patients displayed a 1-year mortality risk > 20%, while about 60% of them carried a 1-year mortality risk below 2%. The adoption of the RAIN-ML model in contemporary practice may allow for the performance of reliable and clinically relevant risk stratification to inform, personalize, and improve care. An accurate risk stratification would allow, on the one hand, targeted strategies in patients with the highest risk of death through a comprehensive evaluation and a tailored approach and, on the other hand, less intensive follow-up for those at low risk.

Despite several differences in the characteristics of the study cohorts, external validation yielded a good performance, suggesting good generalizability and robustness of the RAIN-ML model applicability. In particular, the cohorts of the BIO-RESORT and DUTCH-PEERS trials were composed of patients with a lower burden of cardiovascular risk factors and previous cardiovascular events, translating into lower death rates at follow-up as compared to the real-world clinical setting of the RAIN cohort. Moreover, in the external validation cohorts, “left ventricular ejection fraction”, a powerful feature of the RAIN-ML model, was coded dichotomically, and the variable “diffuse coronary disease” was unavailable. The model performed well despite these missing data and potentially limiting factors that might have affected model discrimination. Importantly, the model performed well at internal validation, suggesting its applicability in the real-world setting and its potential usefulness in daily clinical practice.

### 4.1. Rationale of the Study and Related Work

In this study, we focused on all-cause mortality as the primary endpoint to offer a comprehensive evaluation of the biological risk of this patient subset, which displays unique features as compared to the overall PCI population. Specifically, patients undergoing bifurcation PCI have higher short- and long-term all-cause mortality as compared to patients with non-bifurcation PCI [13,27], pointing to a peculiar association and possibly causal link of bifurcation lesions with mortality. If, on the one hand, the presence of bifurcation lesions is a potential proxy for a more severe atherosclerotic burden, on the other hand, it may also represent a subset biologically more prone to adverse events due to the peculiar rheological characteristics [14,15]. Moreover, specific procedural aspects are associated with long-term mortality in bifurcation PCI, highlighting the importance, beyond the lesion’s natural history, of PCI-related factors in determining the prognosis of this population [5,16]. However, presently there is no predictive tool reflective of current clinical practice available to predict long-term mortality following bifurcation PCI. For these reasons, we focused on a multifaceted approach to residual risk, based on a comprehensive evaluation of patient-, anatomy-, and procedure-related factors integrated by a machine learning approach able to handle multidimensional information and produce data-driven outcome prediction. A key advantage of this approach is that investigators do not generally need to specify which potential predictor variables to consider and in which combinations. A multidimensional approach may be highly relevant in this setting, which is characterized by high anatomical and procedural complexity. Indeed, previous scores to predict adverse events focusing on either patient-related or anatomy/procedure-related factors performed only modestly in patients with bifurcation PCI [28]. Specifically, in a previous analysis of the RAIN registry, the PCI complexity definition proposed by Giustino et al. [11] using validated and guideline-endorsed criteria (reflecting anatomy/procedure-related factors) was unable to discriminate post-procedural mortality (AUC 0.49), and the PARIS risk score [8] (reflecting patient-related factors) displayed only a modest discrimination capacity (AUC 0.65). More importantly, these tools showed potential for an accurate event prediction when combined, thus suggesting that a comprehensive evaluation of clinical, anatomical, and procedural features may better reflect residual risk [28].

Nevertheless, the aim of the RAIN-ML prediction model was not to guide bifurcation PCI based on pre-procedural risk but rather to provide the patient and the treating physician with information that also integrates procedural outcome predictors that may significantly modify the patient’s prognostic trajectory and, consequently, its clinical management and follow-up.

### 4.2. Perspectives

When compared to traditional scores, ML-based models do not always improve performance in terms of accuracy [29]. However, several advantages may become apparent in the long term [30,31,32,33,34,35,36,37]. Specifically, compared to the static nature of traditional scores, the performance of the RAIN-ML prediction model is dynamic, thanks to its evolutive learning feature allowing the model to improve its classification algorithm by learning strategies at the increased enrollment time and number of recruited patients.

An example of continual learning applied to the discovery cohort of the RAIN-ML model is presented in Figure 4: the model was trained at each time point on an increasing number of patients. From 3 months to 33 months of enrollment, the accuracy increased from 67.9% to 78.7% at validation. We thus will plan through a dedicated anonymized system integrated with the freely available online interface (https://rain.hpc4ai.it; accessed on 12 May 2022) to prospectively endorse RAIN-ML training to constantly improve outcome prediction. Moreover, future works should assess whether the integration of more granular features, such as the characteristics of plaque vulnerability evaluated by intravascular imaging, might have potentially improved outcome prediction. Similarly, features of non-cardiovascular comorbidities, which may be relevant to the mortality endpoint in this complex patient population, have not been evaluated in the present work: future studies may integrate these features with the RAIN-ML model to possibly improve its discriminative capability. Finally, prospective validation of the RAIN-ML model in an external real-world population remains desirable.

### 4.3. Limitations

This study has some limitations to be acknowledged. First, the RAIN registry was retrospective. However, the good generalizability demonstrated in the external validation cohort constituted by bifurcation PCI patients from two randomized trials is reassuring. Second, the model performance in the cohorts and at different time points ranged from moderate to good. As discussed, the integration of more granular technical and anatomical features along with data on non-cardiovascular comorbidities might have potentially improved outcome prediction. However, this would have limited the model adoption as these features are not yet routinely assessed in everyday clinical practice.

Third, our ML model requires several input variables that might discourage its use. However, all these variables are generally readily available, and the user-friendly online interface makes risk estimation at the different evaluated time-points an easy and quick procedure.

## 5. Conclusions

The RAIN-ML prediction model represents the first developed tool combining clinical, anatomical, and procedural features through a machine learning approach to predict all-cause mortality among patients undergoing contemporary coronary bifurcation PCI, with robust performance and generalizability for mortality prediction across different clinical scenarios and at different time points. The adoption of the RAIN-ML model has the potential to improve doctor-to-patient communication, patient management, and clinical research.

## Figures and Tables

**Figure 1 jpm-12-00990-f001:**
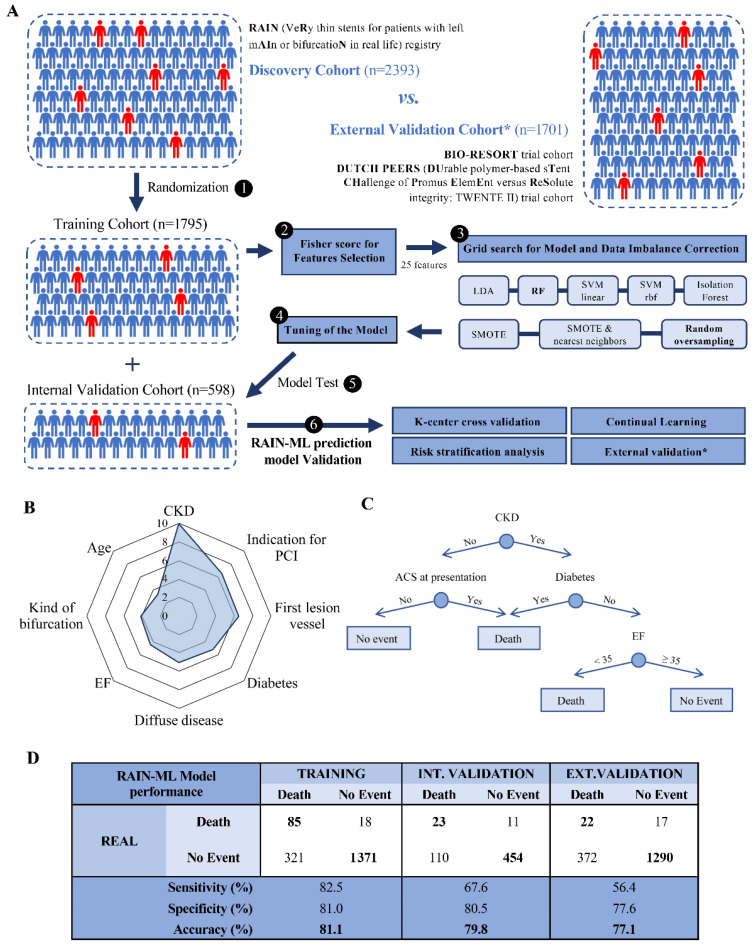
**RAIN-ML model.** The RAIN-ML prediction model was built in the discovery cohort (n = 2393). (**A**) The discovery cohort was randomized in a training and in an internal validation cohort. The model was developed in the training cohort (n = 1795): 5 machine learning models (linear discriminant analysis [LDA], random forest regressor [RF], support vector machine [SVM] with different kernels, and isolation forest) and 3 algorithms for dataset imbalance correction (SMOTE, Synthetic Minority Oversampling Technique, SMOTE & nearest neighbours, and random oversampling) have been evaluated; the best model was an RF with random oversampling algorithm (reported in bold). The model was then tested in the internal and external validation cohorts (n = 598, n = 1701, respectively) and by K-center cross-validation, risk stratification analysis, and continual learning. (**B**) Radar chart reporting the 8 normalized best predictors associated with patient outcome. (**C**) Representative classification tree from the RAIN-ML RF model. (**D**) Confusion matrix, real and predicted diagnosis (Death vs. No event), accuracy, sensitivity, and specificity for the RAIN-ML model at training, internal validation, and external validation. CKD, Chronic Kidney Disease; PCI, Percutaneous Coronary Intervention; EF, Ejection Fraction; ACS, Acute Coronary Syndrome.

**Figure 2 jpm-12-00990-f002:**
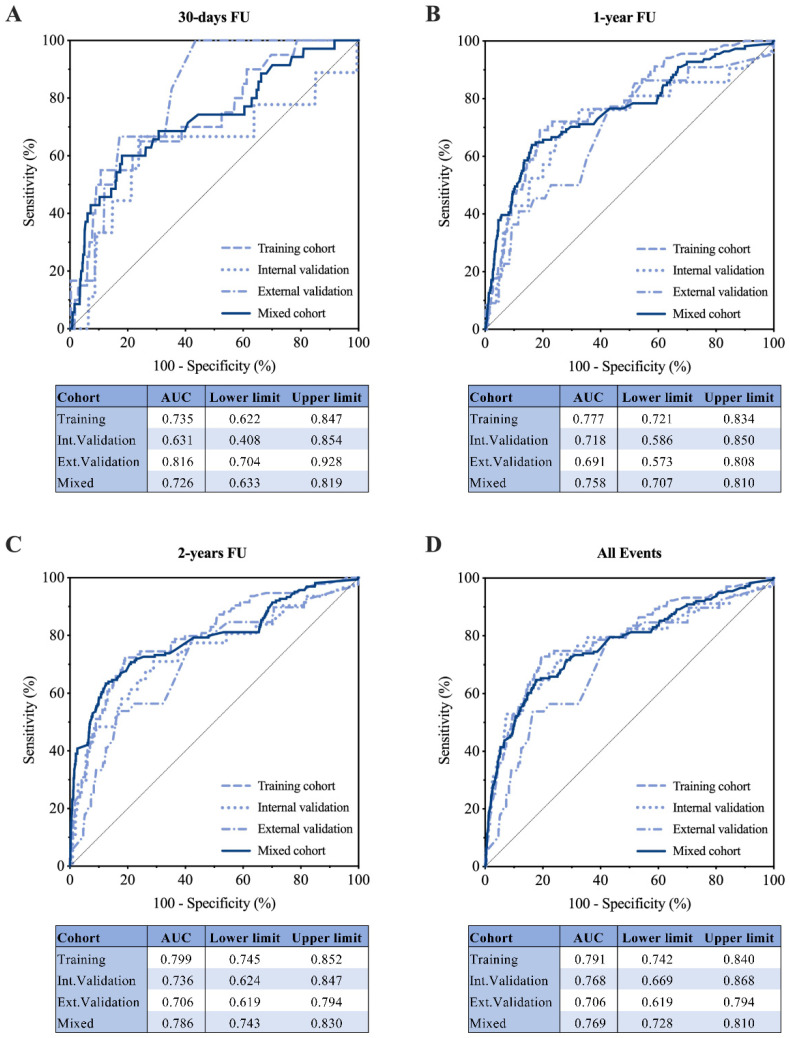
**Predictive performance.** Receiver operating characteristics curve to assess the area under the curve and its 95% confidence interval (lower and upper limits) for the RAIN-ML prediction model at training (n = 1795), internal (n = 598), external validation (n = 1701), and in the mixed cohort (n = 4094). (**A**) Performance at 30-day follow-up; (**B**) Performance at 1-year follow-up; (**C**) Performance at 2-year follow-up; (**D**) Performance considering all the events at follow-up.

**Figure 3 jpm-12-00990-f003:**
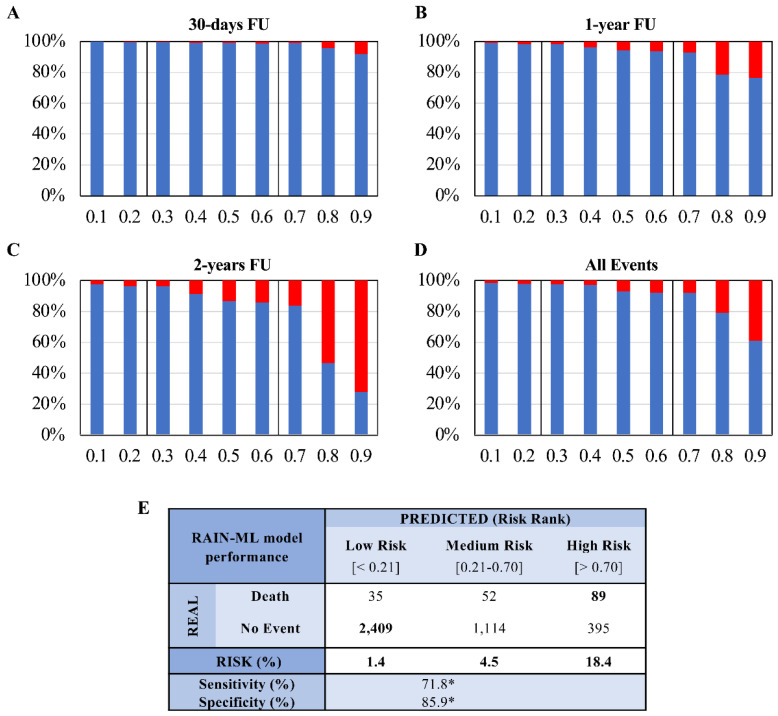
**Stratification of all-cause mortality risk according to the RAIN-ML model**. Patient distribution and risk stratification analysis in the mixed discovery and external validation cohort (n = 4094). (**A**–**D**) Histograms showing the proportion of patients (*y*-axis, %) stratified according to their outcome (No Event, grey vs. Death, black); on the *x*-axis are reported the ML coefficients (for the RAIN-ML prediction model). Patients were stratified considering death occurrence at different follow-ups (30 days, 1 year, 2 years, and all events). (**E**) The table shows confusion matrix reporting risk stratification analysis, sensitivity, and specificity, for the RAIN-ML prediction model * Sensitivity and specificity were derived on a mixed cohort composed of the low- and high-risk groups, after exclusion of patients at intermediate risk.

**Figure 4 jpm-12-00990-f004:**
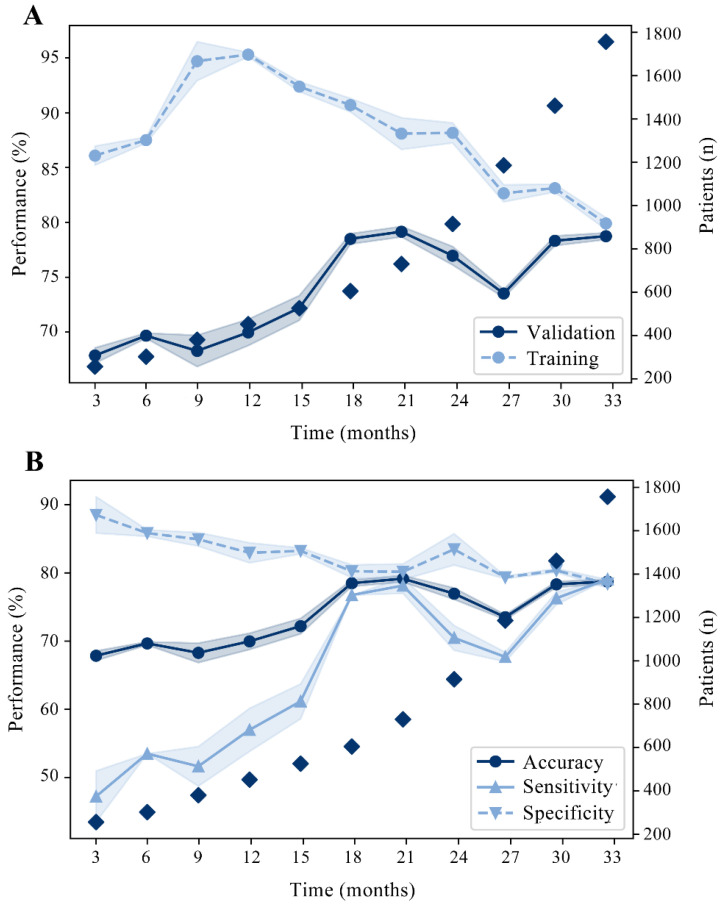
**Continual Learning of RAIN-ML prediction model.** Graphs showing diagnostic performance (left y-axes) with a continual learning strategy in the discovery cohort (n = 2393). Squares indicates the number of patients (right y-axes) over the enrolment time (x-axes). (**A**) Learning simulation for RAIN-ML prediction model at the increase in the enrollment time; 70% of the discovery cohort is used for training (patients enrolled first), 30% for validation (last enrolled patients). Mean and standard deviation are shown after 10 repetitions of the analysis. Accuracy at training: from 86.1% to 79.9%. Accuracy at validation: from 67.9% to 78.7%. (**B**) Accuracy, sensitivity, and specificity at validation.

**Table 1 jpm-12-00990-t001:** Clinical, anatomical, procedural features and outcomes in the study cohorts.

Variable		Training Cohort (n = 1795)	Internal Validation Cohort (n = 598)	External Validation Cohort (n = 1701)
**Patient parameters**	Age (years)	70 [61; 77]	69 [61; 78]	65 [57; 72]
	Sex (ref. male; n, %)	1379 (76.9)	440 (73.6)	1329 (78.1)
	Hypertension (ref. yes; n, %)	1331 (74.2)	460 (76.9)	820 (48.2)
	Hyperlipidemia (ref. yes; n, %)	1074 (59.8)	377 (63.0)	N.A.
	Diabetes (ref. yes; n, %)	605 (33.7)	200 (33.4)	319 (18.8)
	Smoker			N.A.
	No (n, %)	872 (48.6)	291 (48.6)	
	Previous (n, %)	548 (30.5)	181 (30.3)	
	Current (n, %)	375 (20.9)	126 (21.1)	
	CKD (ref. GFR < 60 mL/min; n, %)	391 (21.8)	147 (24.6)	50 (2.9)
	Previous PCI (ref. yes; n, %)	605 (33.7)	187 (31.3)	301 (17.7)
	Previous CABG (ref. yes; n, %)	97 (5.4)	29 (4.8)	120 (7.1)
	Previous MI (ref. yes; n, %)	554 (30.9)	183 (30.6)	339 (19.9)
	EF at echo (%)	55 [55; 60]	55 [55; 65]	50 [50; 50] *
	PCI indication			
	STEMI (n, %)	305 (17.0)	114 (19.1)	383 (22.5)
	NSTEMI (n, %)	447 (24.9)	133 (22.2)	388 (22.8)
	Unstable angina (n, %)	257 (14.3)	90 (15.1)	296 (17.4)
	Stable angina (n, %)	440 (24.6)	147 (24.5)	634 (37.3)
	Positive stress test (n, %)	227 (12.6)	82 (13.7)	0 (0.0)
	Planned angiography (n, %)	119 (6.6)	32 (5.4)	0 (0.0)
	ACS at presentation (ref. yes; n, %)	1007 (56.1)	337 (56.4)	1067 (62.7)
	STEMI at presentation (ref. yes; n, %)	305 (17.0)	114 (91.1)	383 (22.5)
	Kind of DAT (aspirin plus)			
	Clopidogrel (n, %)	1170 (65.2)	391 (65.4)	1131 (66.4)
	Ticagrelor (n, %)	479 (26.7)	162 (27.1)	513 (30.2)
	Prasugrel (n, %)	146 (8.1)	45 (7.5)	57 (3.4)
**Predilatation (ref. yes; n, %)**	First lesion vessel			
	LM (n, %)	435 (24.2)	160 (26.8)	179 (10.5)
	LAD (n, %)	876 (48.9)	296 (49.5)	1044 (61.5)
	Cx/MO (n, %)	320 (17.8)	98 (16.4)	332 (19.5)
	RCA (n, %)	133 (7.4)	33 (5.5)	142 (8.3)
	RI (n, %)	31 (1.7)	11 (1.8)	4 (0.2)
	Lesion localization			
	Ostial (n, %)	64 (3.6)	21 (3.5)	63 (3.7)
	Proximal (n, %)	545 (30.4)	187 (31.3)	964 (56.7)
	Middle (n, %)	642 (35.7)	209 (34.9)	401 (23.6)
	Distal (n, %)	544 (30.3)	181 (30.3)	273 (16.0)
	Type C lesion (ref. yes; n, %)	685 (38.2)	212 (35.5)	N.A.
	Severe calcification (ref. yes; n, %)	261 (14.5)	88 (14.7)	397 (23.3)
	Diffuse disease (ref. yes; n, %)	700 (39.0)	238 (39.8)	N.A.
	Kind of bifurcation			
	Distal LM (n, %)	481 (26.8)	174 (29.1)	179 (10.5)
	LAD/Diag (n, %)	839 (46.7)	284 (47.5)	1045 (61.5)
	Cx/MO (n, %)	341 (19.0)	109 (18.2)	334 (19.6)
	RCA/PL (n, %)	134 (7.5)	31 (5.2)	143 (8.4)
	Medina class			
	1,1,1 (n, %)	605 (33.8)	205 (34.2)	378 (22.2)
	1,1,0 (n, %)	587 (32.7)	198 (33.0)	640 (37.6)
	1,0,1 (n, %)	174 (9.7)	47 (7.9)	85 (5.0)
	0,1,1 (n, %)	87 (4.8)	32 (5.4)	80 (4.7)
	1,0,0 (n, %)	158 (8.8)	41 (6.9)	119 (7.0)
	0,1,0 (n, %)	92 (5.1)	47 (7.9)	258 (15.2)
	0,0,1 (n, %)	92 (5.1)	28 (4.7)	141 (8.3)
	Kind of strategy			
	Provisional (n, %)	1474 (82.1)	490 (81.9)	1447 (85.1)
	Two stents (n, %)	321 (17.9)	108 (18.1)	254 (14.9)
	Use of imaging			
	No (n, %)	1186 (66.1)	400 (66.9)	1648 (96.9)
	IVUS (n, %)	586 (32.6)	194 (32.4)	36 (2.1)
	OCT (n, %)	23 (1.3)	4 (0.7)	17 (1.0)
	Predilatation (ref. yes; n, %)	1559 (86.9)	531 (88.8)	1375 (80.8)
	Kind of balloon			
	Conventional (n, %)	1521 (97.5)	518 (97.5)	1666 (97.9)
	Angiosculpt (n, %)	18 (1.2)	4 (0.8)	35 (2.1)
	Scoring (n, %)	20 (1.5)	9 (1.7)	0 (0.0)
	Rotablator (ref. yes; n, %)	106 (5.9)	38 (6.4)	25 (1.5)
	Kind of stent on MB			N.A.
	Resolute Onyx (n, %)	506 (28.6)	186 (31.3)	
	Xience Alpine (n, %)	454 (25.5)	136 (22.9)	
	Synergy (n, %)	374 (21.0)	131 (22.1)	
	Ultimaster (n, %)	165 (9.3)	50 (8.4)	
	Biomatrix alpha (n, %)	2 (0.1)	2 (0.3)	
	Promus (n, %)	276 (15.5)	89 (15.0)	
	MB lesion diameter (mm)	3.0 [2.8; 3.5]	3.0 [2.8; 3.5]	N.A.
	MB lesion length (mm)	22 [16; 28]	23 [16; 28]	15 [10; 22]
	MB atmospheres (atm)	12 [12; 16]	14 [12; 16]	N.A.
	Stent on SB (ref. yes; n, %)	321 (17.9)	108 (18.1)	442 (26.0)
	SB lesion diameter (mm)	2.3 [1.0; 2.8]	2.5 [1.0; 2.8]	N.A.
	SB lesion length (mm)	28 [20; 33]	28 [20; 33]	N.A.
	SB atmospheres (atm)	12 [10; 14]	12 [12; 14]	N.A.
	POT (ref. yes; n, %)	1384 (77.1)	447 (74.7)	N.A.
	ATM Post (atm)	20 [16; 22]	20 [16; 20]	N.A.
	Final kissing balloon (ref. yes; n, %)	746 (41.6)	248 (41.5)	366 (21.5)
**Outcome**	Death (ref. yes; n, %)	103 (5.7)	34 (5.7)	39 (2.3)
	Median follow-up at the event (days)	274 [61; 433]	253 [23; 458]	284 [65; 500]
	Death within 30 days (ref. yes; n, %)	20 (1.1)	9 (1.5)	6 (0.4)
	Death within 1 year (ref. yes; n, %)	68 (3.8)	21 (3.5)	22 (1.3)
	Death within 2 years (ref. yes; n, %)	94 (5.2)	31 (5.2)	39 (2.3)

Patient and lesion parameters in the discovery cohort (n = 2393) after randomization into training (n = 1795) and internal validation cohorts (n = 598), and in the external validation cohort (n = 1701). Variables are reported as median [interquartile range], or absolute number (percentage, %), as appropriated. CKD, Chronic Kidney Disease; PCI, Percutaneous Coronary Intervention; CABG, Coronary Artery Bypass Graft; MI, Myocardial Infarction; EF, Ejection Fraction; STEMI, ST-segment Elevated Myocardial Infarction; NSTEMI, Non-ST-segment Elevated Myocardial Infarction; ACS, Acute Coronary Syndrome; DAT, Double Antiaggregant Therapy; LM, Left Main; LAD, Left Anterior Descending; Cx/MO, Circumflex/Marginal; RCA, Right Coronary Artery; RI, Right Intermedius; Diag, Diagonal; PL, Posterior Left; IVUS, IntraVascular UltraSound; OCT, Optical Coherence Tomography; MB, Main Branch; SB, Side Branch; POT, Proximal Optimization Technique. *For the external validation cohort, EF was coded dichotomously and reported for each patient as a value of either “30%” or “50%”.

## Data Availability

The data that support the findings of this study are available on reasonable request from the corresponding author.

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
