# Peer review of "Prediction of All-Cause Mortality Following Percutaneous Coronary Intervention in Bifurcation Lesions Using Machine Learning Algorithms"

_jpm, 2022, doi:10.3390/jpm12060990_

Round 1

Reviewer 1 Report

The authors have chosen a very important topic. The paper Is very interesting and timely. I suggest the authors incorporate these changes to further improve the quality of the paper.

1.     There are many Typos and grammatical errors throughout the manuscript.

2.     Figures clarity should be thoroughly enhanced.

3.     The introduction lacks many important references in the field. For example, please include the below citations:

Ambati, Loknath Sai; El-Gayar, Omar; and Nawar, Nevine, "Design Principles for Multiple Sclerosis Mobile Self-Management Applications: A Patient-Centric Perspective" (2021). AMCIS 2021 Proceedings. 11.

Ambati, L. S., El-Gayar, O., & Nawar, N. (2020). INFLUENCE OF THE DIGITAL DIVIDE AND SOCIO-ECONOMIC FACTORS ON PREVALENCE OF DIABETES. Issues in Information Systems21(4), 103-113.

4.     I wish to reorganize your paper with better and more clarity by Rectifying the above comments and submit the revised version.

5.     Introduction and related work is not synchronized.

6.     The conclusion and future work part can be extended to have a better understanding of the approach and issues related to that which can be taken into consideration for future work.

Reviewer 2 Report

Paper is important for health informatics. Following revisions are to be incorporated before publication -

(1)   Author should add the motivations, problem, and solution statement in the abstract.

(2)   How the parameters for simulations are selected?

(3)   How the performance of proposed technique is better than existing techniques like LSTM, CNN, ANN, RVM, etc.

(4)   All tables and figures should be explained clearly.

(5)   The English and typo errors of the paper should be checked in the presence of native English speaker.

(6)   All equations should be clearly explained with explanation on all associated variables.

(7)   Author should add one section “Related Work” in the paper.

(8)   The methodology of the paper should be clearly explained with appropriate flow charts.

(9)   Highlight the more applications of the proposed technique.

(10) What are motivations behind this research work?

(11) Add more explanation on obtained results with critical analysis.

(12) Author must explain pros and cons of the work.

(13) What are the major issues in random forests and LDA?

(14) How over fitting is minimized in the proposed work?

(15) Author must cite suggested papers for enhancing the quality of the paper-

(a)    Relationship between stride interval variability and aging: use of linear and non-linear estimators for gait variability assessment in assisted living environments

(b)   A brain tumor image segmentation technique in image processing using ICA-LDA algorithm with ARHE model

(c)    Neuropathic complications: Type II diabetes mellitus and other risky parameters using machine learning algorithms

(d)   Investigation on automated surveillance monitoring for human identification and recognition using face and iris biometric

(e)    A reinforced random forest model for enhanced crop yield prediction by integrating agrarian parameters

(f)    RL based hyper-parameters optimization algorithm (ROA) for convolutional neural network

(g)   Prediction of atherosclerosis pathology in retinal fundal images with machine learning approaches

(h)   Robust retinal blood vessel segmentation using convolutional neural network and support vector machine

(i)    A kernel support vector machine based anomaly detection using spatio-temporal motion pattern models in extremely crowded scenes

(j)    ECG signal analysis using CWT, spectrogram and autoregressive technique

Round 2

Reviewer 2 Report

Authors have done all recommended corrections. Now paper is acceptable in current form.

Strong Points:
➢ The revised version has come out very well.
➢ Real world problem has been identified and systematically analyzed.
➢ Systematic analysis of the research problem is done with proper block diagrams and tables.
➢ The entire paper is properly divided into sub-sections. Each sub-section has clear introduction, content and final remarks.
➢ Conclusions are in-line with objectives of the research area.

Weak Points:
➢ All weak points have been addressed in revised version.
